# Interpreting adversarial attacks and defences using architectures with enhanced interpretability

## Abstract

Adversarial attacks in deep learning represent a significant threat to the integrity and reliability of machine learning models. These attacks involve intentionally crafting perturbations to input data that, while often imperceptible to humans, can lead to incorrect predictions by the model. This phenomenon exposes vulnerabilities in deep learning systems across various applications, from image recognition to natural language processing. Adversarial training has been a popular defence technique against these adversarial attacks. The research community has been increasingly interested in interpreting robust models and understanding how they defend against attacks.

In this work, we capitalize on a network architecture, namely Deep Linearly Gated Networks (DLGN), which has better interpretation capabilities than regular network architectures. Using this architecture, we interpret robust models trained using PGD adversarial training (9) and compare them with standard training. Feature networks in these architectures act as feature extractors, making them the only medium through which an adversary can attack the model. So, we use the feature network in this architecture with fully connected layers to analyse properties like alignment of the hyperplanes, hyperplane relation with PCA, and sub-network overlap among classes and compare these properties between robust and standard models. We also consider this architecture having CNN layers wherein we qualitatively and quantitatively contrast gating patterns between robust and standard models. We use ideas from visualization to understand the representations used by robust and standard models.

## 1 Introduction and related works

Relu activation can be viewed as the product of input and gates that are off/on. These gates trigger certain pathways in the network to be active/inactive. Lakshminarayanan & Vikram Singh (2020) propose a unique approach by viewing model training as active sub-network learning in Relu-activated neural networks. Neural networks can be viewed as model input being mapped into the path space (path space representation given by neural path features (NPF)) wherein they are combined together in the path space to generate model output logits. The coefficients of these combinations in path space are provided by the model weights, captured using neural path value (NPV). They introduce Deep Gated Neural Network (DGN) architecture to demonstrate the role of active sub-network learning that has two nearly identical sub-networks: *feature network*, which is responsible for extracting features and providing gating signals (thereby solely encoding NPFs); *value network*, which aggregates the features extracted by the feature network (thereby solely encoding NPVs) to produce the final model prediction. A follow-up study by Lakshminarayanan et al. (2022) show that interpreting the value network *visually* is meaningless in DGN networks. However interpreting feature network is still hard due to the non-linearity in the feature network layers. So, to improve interpretability of DGNs, they propose new architecture namely *Deep Linearly Gated Neural Networks (DLGN)* wherein the gating signals are completely moved out of the feature network, rendering the transformations in the feature network entirely linear. The DLGN architecture offer significant interpretability advantages due to the feature network being entirely linear, facilitating understanding and analysis.

Though machine learning algorithms perform well under normal conditions, they can fail with cleverly crafted inputs called adversarial samples, raising security concerns in many applications. White-box attacks are attacks wherein the attacker can access model predictions, parameters and training data. Popular attacks in this setting are BIM(Kurakin et al. (2016)), MIM(Dong et al. (2018)), FGSM(Goodfellow et al. (2015)) and PGD(Madry et al. (2017)) among which PGD attacks are considered one of the strongest white-box attacks in practice. Prior works have proposed various defence techniques against adversarial attacks, among which the seminal work of Madry et al. (2017) stands out as one of the principled methods. They view defending adversarial attacks as solving a min-max optimization problem wherein the inner maximization aims to get the best possible adversarial samples at a given model state. They solve the inner maximization by using the PGD attack and call it *adversarial training* (Algorithm 1) (abbreviated as PGD-AT henceforth). This arms race between adversarial attacks and defenses has also lead to many works which instead analyse the adversarial attacks in several ways like distribution shift analysis3, Fourier spectrum analysis (8, 11, 13, 12, 10), principal component (analysis3), shapely value analysis (1) and so on.

---

**Algorithm 1** PGD adversarial training for $M$ epochs, given some radius $\epsilon$, adversarial step size $\alpha$, $T$ PGD steps and a dataset of size $N$ for a network $F_\theta$

---

   **for** $j = 1 \ldots M$ **do**
     **for** $i = 1 \ldots N$ **do**
       *// Perform PGD adversarial attack*
       $\delta = U(-\epsilon, \epsilon)$
       **for** $t = 1 \ldots T$ **do**
         $\delta = \delta + \alpha \cdot \text{sign}(\nabla_\delta L(F_\theta(x_i + \delta), y_i^{true}))$
         $\delta = \max(\min(\delta, \epsilon), -\epsilon)$
       **end for**
       $\theta = \theta - \nabla_\theta L(F_\theta(x_i + \delta), y_i)$ *// Update model weights with some optimizer, e.g. SGD*
     **end for**
   **end for**

---

We use the enhanced interpretation capabilities of DLGN model (see Appendix A for network architecture) to compare and contrast standard training (henceforth abbreviated as STD-TR) and adversarial training by analysing the model's internals, which was previously challenging in traditional architectures due to the non-linearity between the layers.

**Our Contributions**

- We merge layers in the feature network of DLGN architectures to obtain a single effective linear transformation per layer. This reveals novel insights into hyperplanes and their resemblance to principal components in PGD-AT and STD-TR models. Our analyses show that hyperplanes in PGD-AT (FC) models are farther from data points compared to STD-TR (FC) models and play a key role in enhancing robustness.

- We analyze path activity among classes by examining the active-subnetwork overlap in PGD-AT and STD-TR FC models. Our findings indicate that PGD-AT models generate more diverse active subnetworks and can avoid active subnetwork overlaps with different classes during an attack.

- We quantitatively compare active gate overlaps among classes using the intersection-over-union metric. This reveals that adversarially trained models can prevent significant gating pattern changes and avoid overlap of attack-induced gating changes with those of other classes. Using feature inversion visualization techniques, we interpret the representations used by PGD-AT and STD-TR models.

**Notations** *The following are the notations in fully connected architectures:* Let $\theta_f$ and $\theta_v$ be parameters of the model with $L$ layers in feature network and value network respectively and more specifically with $W_l \in R^{m_{l-1}, m_l}$ being the weight at layer $l$ of feature network, $b_l \in R^{m_l}$ being the bias at layer $l$ of the feature network. Let $x_l \in R^{m_l}$ be the feature network output at layer $l$, $p$ be one of the paths among total $P$ paths passing from each input node to each output node, $G_{x,\theta_f}^{l,p}$ be the gate for input $x$ at the node contained in path $p$ at layer $l$ and $x_p$ be the input node at node contained in path $p$. Then from work [6], the gate information is encoded in the neural path features (NPF)

$\Phi_{x,\theta_f} \in R^P$ as per Equation (1b) and the weight information is encoded in the neural path value (NPV) $\vartheta_{\theta_v} \in R^P$ as per Equation (1c). The final model output logits is given by $\hat{y}(x)$, which can be expressed as per Equation (1d).

$$G^l_{x,\theta_f} = \sigma(\beta * (W_l^T x_{l-1} + b_l)) \tag{1a}$$

$$\Phi_{x,\theta_f} = \{x_p \Pi_{l=1}^L G^{l,p}_{x,\theta_f}, p \in [P]\} \in R^P \tag{1b}$$

$$\vartheta_{\theta_v} = \{\Pi_{l=1}^L \theta_v^{l,p}, p \in [P]\} \in R^P \tag{1c}$$

$$\hat{y}(x) = <\Phi_{x,\theta_f}, \vartheta_{\theta_v}> \tag{1d}$$

where $\sigma$ is the sigmoid activation i.e $\sigma(x) = \dfrac{1}{1 + e^{-x}}$

## 2 ANALYSIS OF HYPERPLANES IN FEATURE NETWORK OF FULLY CONNECTED ROBUST AND STANDARD MODELS

Consider a DLGN architecture with fully connected layers where the feature network is entirely linear. At each *feature network layer* $l$, the effective linear transformation can be obtained by merging all preceding layers up to $l$, with effective weights $E_l \in R^{m_0, m_l}$ and bias $p_l \in R^{m_l}$. The output at layer $l$ would produce $m_l$ gates and each gate's effective weight $\in R^{m_0}$ would be a hyperplane acting on input in $m_0$-dimensional space. A gate is active/inactive based on which side of the hyperplane the input $x$ lies.

$$\hat{y}(x + \delta) = \sum_{p=1}^P \Phi_{x+\delta,\theta_f} * \vartheta_{\theta_v} = \sum_{p=1}^P [(x^p + \delta^p)\Pi_{l=1}^L \sigma\{E_l^p(x + \delta) + p_l)\}] * \vartheta_{\theta_v}$$

$$\hat{y}(x + \delta) = \sum_{p=1}^P [(x^p + \delta^p)\Pi_{l=1}^L \sigma\{E_l^p x + E_l^p \delta + p_l)\}] * \vartheta_{\theta_v} \tag{2a}$$

From Equation (2a) for a perturbation $\delta$ in input $x$, larger values of $E_l^p x + p_l$ reduce the gate's sensitivity in path $p$, enhancing robustness by preventing changes in model output $\hat{y}$ for perturbed inputs. Informally, if a point is farther from a hyperplane, it requires either larger dimension-wise perturbations or small perturbations across many dimensions to flip the gate[1].

### 2.1 HYPERPLANE ANALYSIS IN REAL-WORLD DATASET

We trained a DLGN with 4 fully connected layers (width 128) on the MNIST and Fashion MNIST datasets using both standard training and adversarial training (PGD-AT, $\epsilon = 0.3, \alpha = 0.1, T = 40$). Adversarial attacks used PGD with 40 steps and $\epsilon = 0.3$. When an adversarial example crosses to the opposite side of the hyperplane compared to the original input, the gate is considered flipped (from active to inactive or vice versa). As shown in Figure 1, fewer data points flipped at each hyperplane in PGD-AT models than in STD-TR models. We inspect the projection distance of points from each hyperplane of fully connected layers in the feature network of DLGN given by the expression $\frac{E_l^T x + p_l}{||E_l||_2}$. Guided by the mathematical intuition at Equation (2a), we experimentally (in Figure 2 & Figure 3) show that larger median projection distances results in less gate flipping thereby enhancing robustness. We plot the median projection distance over all samples from each hyperplane across all layers (see Figure 4 and Appendix A.1) and found that the median distance from hyperplane is relatively higher in PGD-AT models than STD-TR models at many hyperplane indices. This trend is also reflected in projection distance histograms, which show significant differences between standard and robust models (see Appendix A.1). We compare masking gates with the highest median projection distance, masking gates with lowest median projection distance and masking gates randomly in PGD-AT and STD-TR models (see Figure 5). Results show that median-based masking significantly reduces PGD-40 and clean accuracies in PGD-AT models, highlighting the importance of gates with higher median distances for robustness.

---

[1]In experiments, $\beta$ is set high to approximate a step function

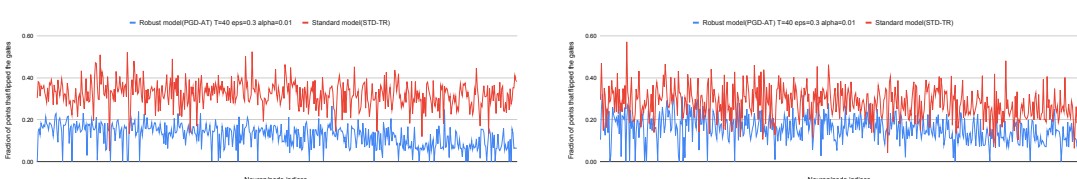

Figure 1: PGD-AT vs STD-TR FC-DLGN-W128-D4 flip distribution. The left image is on MNIST, and the right image is on the Fashion MNIST dataset. The Y-axis denotes the fraction of points that flipped the gate at node indices on the X-axis.

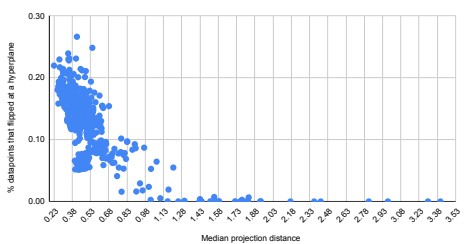

Figure 2: PGD-AT per hyperplane flip distribution vs. median projection distance

Figure 3: STD-TR per hyperplane flip distribution vs. median projection distance

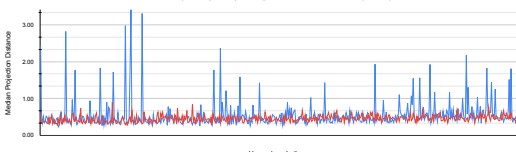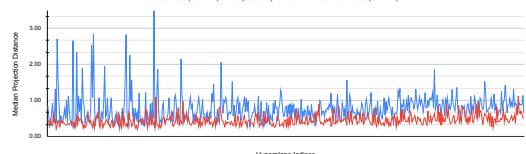

Figure 4: PGD-AT vs STD-TR FC-DLGN-W128-D4 median projection distance. The left image is on MNIST, and the right image is on the Fashion MNIST dataset. The Y-axis denotes the median projection distance of data points at node/hyperplane indices on the X-axis.

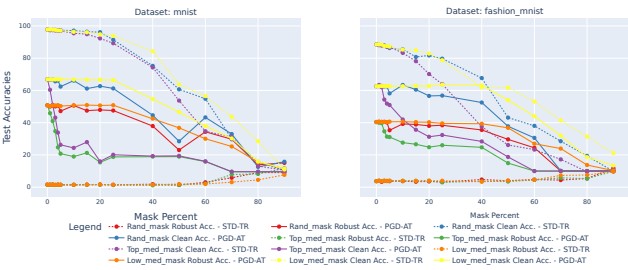

Figure 5: Robust and clean accuracies of PGD-AT and STD-TR FC-DLGN _W128_D4 models with random gate masking vs. masking gates with the highest median projection distance vs masking gates with lowest median projection distance. Dotted lines are for STD-TR models and solid lines are for PGD-AT models.

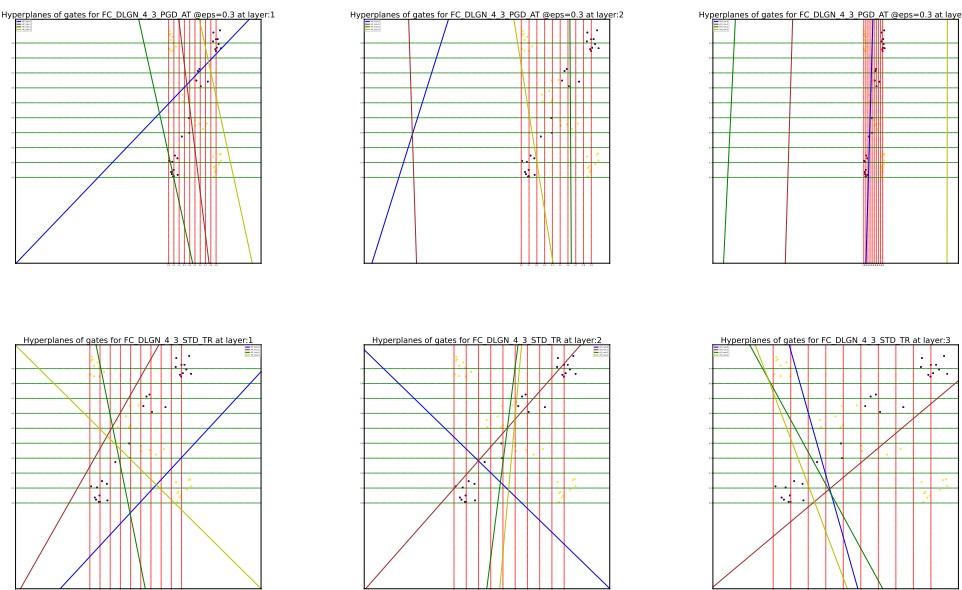

Figure 6: Hyperplane plots of PGD-AT vs STD-TR models in FC-DLGN-W4-D3. Row 1 indicates the PGD-AT model, and row 2 indicates the STD-TR model. Columns 1-3 indicate layers 1-3. Each image contains 4 hyperplanes since width at each layer is 4.

## 2.2 HYPERPLANE ANALYSIS IN SYNTHETIC XOR DATASET

We constructed a 2D XOR dataset (see Appendix A.2) with a gap $\lambda$ from the axes, ensuring that points satisfy $|x| > \lambda$ and $|y| > \lambda$. This design allows setting $\epsilon \leq \lambda$ during adversarial training, where $\epsilon$ represents the perturbation boundary without changing the ground truth labels. Using a DLGN with 3 fully connected layers (width 4), we trained models via both standard (STD-TR) and adversarial training (PGD-AT, $\epsilon = 0.3, T = 40$). The decision boundaries of PGD-AT models are closer to optimal compared to STD-TR (see Appendix A.2), ensuring that adversarial examples within $L_\infty$ bounds ($\epsilon = 0.3$) are correctly classified only by PGD-AT. Visualization of hyperplanes at each layer of the feature network (see Figure 6) shows that PGD-AT models learn hyperplanes positioned farther from the data points than STD-TR models. This trend increases in deeper layers as compared to earlier ones. So, we conclude that hyperplanes with larger projection distances from data points are key in enhancing robustness.

## 3 PCA ANALYSIS IN ROBUST AND STANDARD MODELS

Principal Component Analysis (PCA) minimizes point-to-hyperplane distances, while we saw that the PGD-AT process increases these distances to improve robustness. This fundamental difference motivates us to investigate the impact of PCA on adversarially training. We *embedded* PCA projection operation into the input layer of a DLGN architecture, ensuring both training and inference accounted for the transformation. This also ensures that adversary has knowledge of the operation and doesn't change dimensions of the model input. To offset the reduced capacity from PCA's dimensionality reduction, we increased the model's width at all layers to keep the capacity constant across all models under comparision. Experiments on MNIST and Fashion MNIST (see Figure 7) reveal a significant drop in both PGD-40 and clean accuracy in PGD-AT models compared to STD-TR models, indicating that PCA negatively affects adversarial robustness. This suggests that PCA's dimensionality reduction conflicts with the robustness objectives of adversarial training.

To further investigate the relation of principal components with hyperplanes in PGD-AT models, we computed the top $k$ principal components $P \in \mathbb{R}^{m_0 \times k}$ of the MNIST and Fashion MNIST training datasets and analyzed their similarity with the effective weights $E_l \in \mathbb{R}^{m_0 \times m_l}$ of the

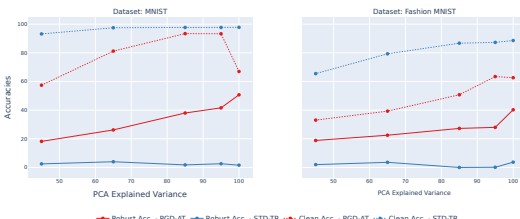

Figure 7: DLGN model trained with PCA embedded layer at different levels of dimensionality reduction on MNIST and Fashion MNIST datasets.

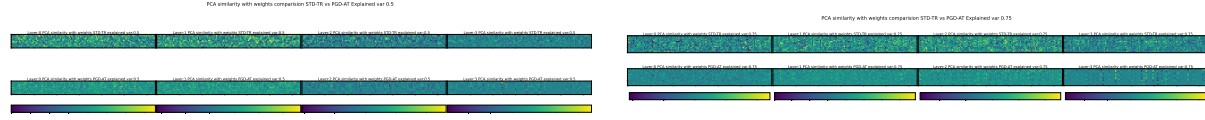

MNIST dataset with explained variance 0.5, 12 components.

Fashion MNIST dataset with explained variance 0.75, 15 components.

Figure 8: Effective weights with top PCA components in PGD-AT(bottom row) and STD-TR(top row) using FC-DLGN-W128-D4 architecture.

feature network layers in models, given by $C_l = P^T E_l \in \mathbb{R}^{k \times m_l}$. Results show higher alignment between principal components and hyperplanes in STD-TR models compared to PGD-AT models (see Figure 8 and Appendix A.3). This supports the observation that PGD-AT hyperplanes are positioned to maximize robustness rather than minimize point-to-hyperplane distance, leading to lower similarity with principal components.

## 4 ACTIVE SUBNETWORK OVERLAP IN FULLY CONNECTED ROBUST VS STANDARD MODELS

Adversaries can alter the output only by changing the active pathways (i.e., NPF). Due to this significance, we measure the overlap in active pathways among samples of the same class and between different classes. The Neural Path Kernel (NPK) $\Psi$ (as per Equation (3a)) is the gram-matrix of NPFs that measures the overlap of active pathways between pairs of examples. We consider a binary classification task and define two metrics to measure overall NPK overlap between different classes $\Psi^D$ and between the same classes $\Psi^S$ as defined in Equation (4).

$$\Psi_\theta(s, s^`) = < \Phi_{x_s, \theta}, \Phi_{x_{s^`}, \theta} > \qquad s, s^` \in [n]\} \in R^{n,n} \qquad (3a)$$

where $\theta$ is parameters of the model, $\Phi_{x, \theta} \in R^P$ is the NPF

$$\Psi^S = \sum_{i,j} \Psi_\theta(i, j) \ \forall i, j : y^i_{true} = y^j_{true} \qquad \Psi^D = \sum_{i,j} \Psi_\theta(i, j) \ \forall i, j : y^i_{true} \neq y^j_{true} \qquad (4)$$

We obtain these two metrics among adversarial ($\Psi_{adv}$), original samples ($\Psi_{orig}$) and between adversarial and original samples ($\Psi_{adv,org}$) for models trained using PGD-AT and STD-TR on two class datasets (see Table 1 for MNIST dataset and Appendix A.4 for Fashion MNIST dataset). Firstly, $\Psi^D_{orig} < \Psi^D_{adv}$ & $\Psi^D_{orig} < \Psi^D_{adv,org}$ for both PGD-AT and STD-TR models. This indicates that adversarial attacks increase active subnetwork overlap between different classes as compared to original samples in an attempt to change the model prediction. Secondly, $\Psi^D_{adv,org}$ for PGD-AT is always lesser than $\Psi^D_{adv,org}$ for STD-TR models. Also $\Psi^D_{adv}$ for PGD-AT is lesser than $\Psi^D_{adv}$ for STD-TR models in most cases. These indicate that the active pathways triggered by adversarial examples overlap less with original examples or adversarial examples of another class in the PGD-AT model. Thirdly, $\Psi^S_{adv}$ & $\Psi^S_{orig}$ for PGD-AT is always lesser than that in STD-TR. So, the trends so far indicate that the PGD-AT training process learns to map the input to a more diverse path space where overlap among the same class is lesser and PGD-AT models control subnetwork overlap between different classes during an attack compared to STD-TR models.

| Dataset | Train Type | PGD-40 Acc. | Clean Acc. | $log_2 \Psi_{orig}^D$ | $log_2 \Psi_{orig}^S$ | $log_2 \Psi_{adv}^D$ | $log_2 \Psi_{adv}^S$ | $log_2 \Psi_{adv,or}^D$ | $log_2 \Psi_{adv,or}^S$ |
|---|---|---|---|---|---|---|---|---|---|
| MNIST | PGD-AT | 76.4% | 83.8% | **24.8** | **26.9** | **25** | **26.4** | **24.8** | **26.1** |
| 3vs8 | STD-TR | 0% | 99.2% | 24.6 | 27.3 | 27.1 | 28 | 26.9 | 26.2 |
| MNIST | PGD-AT | 90.6% | 95.5% | **22.8** | **27.3** | **25** | **26.9** | **23.9** | **26.3** |
| 1vs7 | STD-TR | 0.7% | 99.7% | 20 | 27 | 27.3 | 28 | 25.8 | 26.2 |
| MNIST | PGD-AT | 79.5% | 86.5% | **22.9** | **27.1** | **24.7** | **26.4** | **24** | **26.3** |
| 0vs6 | STD-TR | 0% | 99.4% | 21.2 | 27.6 | 27.8 | 28.4 | 26.5 | 26.3 |
| MNIST | PGD-AT | 85.8% | 94.7% | **22.9** | **26.9** | **25** | **26.4** | **24.1** | **26.1** |
| 1vs5 | STD-TR | 0.4% | 99.8% | 20.42 | 27.5 | 21 | 28.4 | 27.2 | 20.7 |
| MNIST | PGD-AT | 78.2% | 84.5% | **24.4** | **26.8** | **24.6** | **26.2** | **24.4** | **26** |
| 3vs9 | STD-TR | 0% | 99.5% | 23.7 | 27.4 | 26 | 28.3 | 27 | 25 |
| MNIST | PGD-AT | 81.4% | 86.7% | **24** | **27.1** | **24.6** | **26.7** | **24.2** | **26.5** |
| 2vs9 | STD-TR | 0% | 99.6% | 23.3 | 27.3 | 23.5 | 28.5 | 27 | 23.5 |

Table 1: FC-DLGN-W128-D4 architecture PGD-AT vs STD-TR model subnetwork overlap metrics over original and adversarial examples. The task is binary classification over the MNIST dataset in column 1, and the model has a single output node for classification. PGD-AT rows are highlighted in bold for better readability.

**Notations** *The following are the notations in convolutional architectures:* Let $X \in R^{N,1,W,H}$ be the whole training dataset with the size of each sample being $1 \times W \times H$. Let $X_c \in R^{N_c,1,W,H}$ be the training dataset per class with $N_c$ being the number of samples of class $c$. Let $L$ be the number of layers, $C_l$ be the number of output channels in layer $l$ of feature network (in our experiments for simplicity, we keep $C_l$ same across all layers) and $W, H$ be the width, height of output at all feature network layers (since we fix padding=1, kernel size=3, width and height of the output stays same across all layers). Let the output at each feature network layer per class be $F_l \in R^{N_c,C_l,W,H}$. For original examples, let the output combined across all feature network layers be $F^{orig} \in R^{L,N_c,C_l,W,H}$ and for adversarial examples let it be $F^{adv} \in R^{L,N_c,C_l,W,H}$. "mode" is either adversarial or original examples throughout the paper.

# 5 ANALYSIS AND INTERPRETATION OF GATING PATTERNS IN ROBUST VS STANDARD MODELS IN CONVOLUTIONAL ARCHITECTURES

The gates generated in the feature network are the only input representations accessible to the model's value network; hence, their study throws light on the behaviour of robust models.

## 5.1 ANALYSIS OF GATING PATTERNS IN ROBUST AND STANDARD MODELS

Our goal is to measure the extent of active gate overlap among different class-pairs in *convolutional* DLGN architectures quantitatively using the idea of intersection-over-union(IOU) and qualitatively by visually inspecting the difference in active gate counts with and without attacks (refer Appendix A.5). The number of active gates per class at each pixel in $F_l$ across all $L$ layers is given by Equation (5).

$$Gate(x) = \begin{cases} 1, & \text{if } x > 0 \\ 0, & \text{otherwise} \end{cases} \qquad \Lambda_c^{mode} = \sum_{i=1}^{N_c} Gate(F^{mode}(X_c)), \in R^{L,C_l,W,H} \qquad (5)$$

The following is the procedure to obtain IOU of active gate count of class $c_1$ and $c_2$ ($IOU_{agc}(c_1, c_2)$):

1. Compute union of active gate counts at all pixels
   $A_{c_1,c_2}^{mode} : A_{c_1,c_2}^{mode}(i) = \Lambda_{c_1}^{mode}(i) + \Lambda_{c_2}^{mode}(i) \qquad \forall i \in R^{L,C_l,W,H}$

2. Compute intersection of active gate counts at all pixels
   $B_{c_1,c_2}^{mode} : B_{c_1,c_2}^{mode}(i) = min(\Lambda_{c_1}^{mode}(i), \Lambda_{c_2}^{mode}(i)) \qquad \forall i \in R^{L,C_l,W,H}$

3. Record the indices of $U_{c_1,c_2}^{mode}$ whose value is such that $A_{c_1,c_2}^{mode}(i) > 0.1 * (|X_{c_1}| + |X_{c_2}|)$. Let such an index vector be $\iota_{c_1,c_2} \in R^d$. The intent of this stage is to remove outliers in the union of active gate counts of both classes.

| Src Class | Train Type | Quantity | Class 0 | Class 1 | Class 2 | Class 3 | Class 4 | Class 5 | Class 6 | Class 7 | Class 8 | Class 9 |
|---|---|---|---|---|---|---|---|---|---|---|---|---|
| 0 | PGD-AT | $IOU_{agc}^{adv}$ | 100 | 70.2 | 83 | 82.7 | 81.8 | 83.9 | 82.7 | 77.4 | 84 | 80.6 |
| | | $IOU_{agc}^{org}$ | 100 | 66.2 | 79.3 | 79.4 | 77.9 | 81.3 | 78.5 | 72.7 | 81.2 | 76 |
| | STD-TR | $IOU_{agc}^{adv}$ | 100 | 78.1 | 84.7 | 82 | 81 | 84.5 | 84 | 80 | 78.2 | 82.6 |
| | | $IOU_{agc}^{org}$ | 100 | 59.7 | 74.7 | 75 | 73.1 | 77.6 | 73.2 | 66.2 | 77.5 | 69.7 |
| 1 | PGD-AT | $IOU_{agc}^{adv}$ | 70.2 | 100 | 74.9 | 75.8 | 74.7 | 74.2 | 75 | 77.2 | 76.3 | 75.6 |
| | | $IOU_{agc}^{org}$ | 66.2 | 100 | 71.9 | 74.3 | 71.9 | 71 | 71.6 | 75 | 74.2 | 73.6 |
| | STD-TR | $IOU_{agc}^{adv}$ | 78.1 | 100 | 82.7 | 79.5 | 80 | 80 | 79.3 | 83.3 | 74.9 | 80.4 |
| | | $IOU_{agc}^{org}$ | 59.7 | 100 | 63.7 | 66.5 | 65.6 | 64.4 | 64.7 | 67.4 | 68.4 | 67.2 |
| 2 | PGD-AT | $IOU_{agc}^{adv}$ | 83 | 74.9 | 100 | 86.9 | 84.3 | 83.5 | 85.8 | 79.7 | 86.1 | 82.4 |
| | | $IOU_{agc}^{org}$ | 79.3 | 71.9 | 100 | 84.7 | 80.5 | 80.1 | 82.7 | 74.9 | 83.4 | 78.3 |
| | STD-TR | $IOU_{agc}^{adv}$ | 84.7 | 82.7 | 100 | 82.4 | 83.9 | 83.3 | 85.7 | 82 | 79.2 | 83 |
| | | $IOU_{agc}^{org}$ | 74.7 | 63.7 | 100 | 80.7 | 74.2 | 74.9 | 77 | 68.2 | 77.5 | 70.5 |
| 3 | PGD-AT | $IOU_{agc}^{adv}$ | 82.8 | 75.8 | 86.9 | 100 | 82.7 | 86.3 | 82.1 | 81.7 | 87.4 | 83 |
| | | $IOU_{agc}^{org}$ | 79.4 | 74.3 | 84.7 | 100 | 78.6 | 84.7 | 78 | 77.8 | 85.4 | 79 |
| | STD-TR | $IOU_{agc}^{adv}$ | 82 | 79.5 | 82.4 | 100 | 77.5 | 85.4 | 77.6 | 83.6 | 75.7 | 81.8 |
| | | $IOU_{agc}^{org}$ | 75 | 66.5 | 80.7 | 100 | 73.4 | 80.8 | 72.3 | 71.9 | 81.4 | 73 |
| 4 | PGD-AT | $IOU_{agc}^{adv}$ | 81.8 | 74.7 | 84.3 | 82.7 | 100 | 85.8 | 84.7 | 85.4 | 86.9 | 91.1 |
| | | $IOU_{agc}^{org}$ | 77.9 | 71.9 | 80.5 | 78.6 | 100 | 82.3 | 81.3 | 82.8 | 84.5 | 90.6 |
| | STD-TR | $IOU_{agc}^{adv}$ | 81 | 80.2 | 83.9 | 77.8 | 100 | 81.2 | 82.5 | 80.7 | 80.8 | 87.2 |
| | | $IOU_{agc}^{org}$ | 73.1 | 65.6 | 74.2 | 73.4 | 100 | 78.5 | 75 | 77.7 | 80 | 85.9 |

Table 2: CONV DLGN-N128-D4 PGD-AT vs STD-TR model IOU of active gate count between class-pairs over adversarial and original examples in MNIST dataset.

4. Obtain the final intersection as $I_{c_1,c_2}^{mode} = B_{c_1,c_2}[\iota_{c_1,c_2}^{mode}] \in R^d$. Obtain the final union region as $U_{c_1,c_2}^{mode} = A_{c_1,c_2}^{mode}[\iota_{c_1,c_2}^{mode}] \in R^d$

5. Obtain overall average IOU between classes $c_1, c_2$ as

$$IOU_{agc}^{mode}(c_1, c_2) = \frac{1}{d} \sum_{i=1}^{d} \frac{I_{c_1,c_2}^{mode}(i)}{U_{c_1,c_2}^{mode}(i)}$$

We trained a DLGN with 4 convolutional layers, each having 128 filters (padding 1, stride 1, kernel size 3), followed by an adaptive average pooling layer and a fully connected classification layer. Adversarial training (PGD-AT) was performed on the MNIST dataset with $\epsilon = 0.3, T = 40, \alpha = 0.005$, and we measured the Intersection-over-Union (IOU) of active gate overlaps between different class pairs over original ($IOU_{agc}^{org}$) and adversarial ($IOU_{agc}^{adv}$) samples (see Table 2 for MNIST and Appendix A.6 for Fashion MNIST). First, $IOU_{agc}^{org}$ for PGD-AT models is consistently higher than for STD-TR models, indicating that gate overlap among classes is initially larger in PGD-AT models. Second, for both PGD-AT and STD-TR, adversarial attacks increase the gate overlap, as $IOU_{agc}^{adv} > IOU_{agc}^{org}$ across all class pairs. Third, the increase in gate overlap ($IOU_{agc}^{adv} - IOU_{agc}^{org}$) is larger in STD-TR models compared to PGD-AT models, demonstrating that minimizing gate overlap among different classes during adversarial attacks is a key feature of PGD-AT models.

## 5.2 INTERPRETATION OF GATING PATTERNS IN ROBUST VS STANDARD MODELS

We aim to further analyze gating patterns by identifying the images that most effectively trigger them. We begin by inverting gating signals in the DLGN model trained in both PGD-AT and STD-TR modes. Then, we explore more complex gating patterns through inversion. We start by asking: *What is the input image that best simulates the dominant gating signals of an entire class?*. The procedure to obtain such an input image $I$ for class $c$ is as follows:

1. Obtain the active gate count per pixel $\Lambda_c^{mode}$ as per Equation (5). Also obtain the inactive gate count per pixel $\eta_c^{mode} : \eta_c^{mode}(i) = N_c - \Lambda_c^{mode}(i)$.

2. Obtain the dominating gate active-inactive trend per pixel $\rho_{c,\lambda}^{mode}$ as per Equation (6). Here $\lambda$ is the threshold which indicates the percentage of gates that has to be active(inactive) among

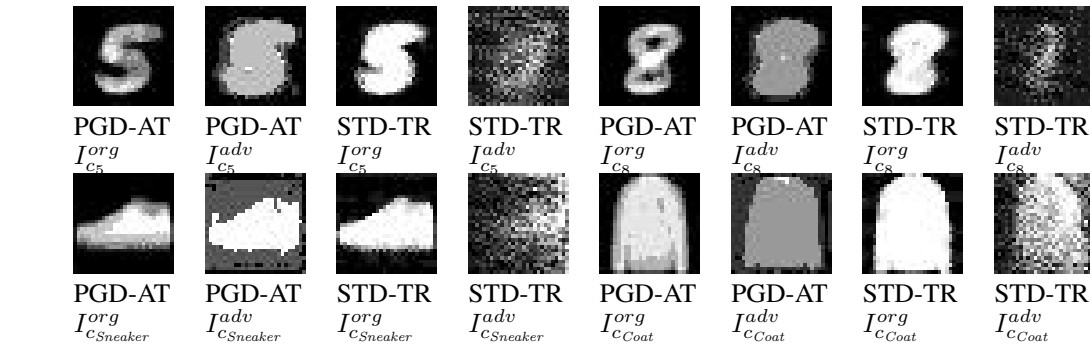

Figure 9: Image $I$ which triggers dominating gating pattern per class over CONV DLGN-N128-D4 model obtained on adversarial examples (even columns) and original examples (odd columns). Columns 1,2,5,6 are on the PGD-AT model, and columns 3,4,7,8 on the STD-TR model. Visualization loss function is as per Equation (7),$\lambda = 0.9$, $\alpha = 0.1$,optimization is as per Equation (8).We have reported a few classes for brevity. Detailed results are in Table 8.

all the class samples to be considered as active(inactive) overall.

$$\rho_{c,\lambda}^{mode}(i) = \begin{cases} 1, iff\Lambda_c^{mode}(i) > \lambda * N_c \\ -1, iff\eta_c^{mode}(i) > \lambda * N_c \end{cases} \qquad (6)$$

3. Let $I$ be the input image under optimization, $F^{mode}$ be the feature maps at the feature network for input $I$ as usual as per our notations. Then, we define a loss function $L(I, \rho^{mode})$ as per Equation (7). This loss function objective is to obtain $I$ such that its feature maps sign at each pixel matches with the dominating gate pattern.

$$L(I, \rho^{mode}) = \sum_i log(1 + e^{-\rho(i)*tanh(F(i))}) \qquad (7)$$

4. Now we need to optimize $I$ over the loss function. We explored gaussian blur on gradient and $I$ route but found the results to be satisfactory. However we found the optimization mentioned in Equation (8) provides good results.

$$I_t = I_{t-1} + \alpha sign(\nabla_{I_t} L) \qquad (8)$$

5. Start with $I_0 = 0$ and perform optimization as per Equation (8) on the loss function Equation (7) for T steps. That is, repeat step 3,4 T times.

In our experiments, we set $\alpha = 0.1$, $T = 50$, and $\lambda = 0.9$. The visualizations for DLGN _N128_D4 trained on the MNIST, Fashion MNIST dataset are presented in Figure 9. Dominant gating patterns from original examples capture critical class information, with images inverting these patterns ($I^{org}$) clearly resembling their respective classes. The PGD-AT model produces sharper, more distinct class features than the STD-TR model, indicating better utilization of model capacity by PGD-AT. In the STD-TR model, for example, $I_5^{org}$ can be made to resemble $I_8^{org}$ with less changes, indicating that one can change input image that triggers dominant gates of class 5 to the image that triggers dominant gates of class 8 easily, thereby showing the brittle nature of representations used by STD-TR models. Furthermore, in PGD-AT, $I^{adv}$ retain class resemblance, while STD-TR's $I^{adv}$ images are noisy. This indicates PGD-AT prevents adversaries from activating semantically unrelated gating patterns, maintaining class information with slight degradation.

Next, we aim to find the input images ($I^{ado}$) that best simulate gating signals dominantly active during adversarial attacks but not in original examples for an entire class and the images ($I^{amo}$) that simulate gate patterns active in both adversarial and original examples for an entire class. The visualization process remains the same, except for changes in the computation of $\rho$. $I_c^{ado}$ is derived using $\rho_c^{ado}$ , while $I_c^{amo}$ uses $\rho_c^{amo}$ as per Equation (9).

$$\rho_{c,\lambda}^{ado}(i) = Gate\{\rho_{c,\lambda}^{adv}(i) - Gate(\rho_{c,\lambda}^{org}(i))\} \qquad \rho_{c,\lambda}^{amo}(i) = \rho_{c,\lambda}^{adv}(i) * Gate(\rho_{c,\lambda}^{org}(i)) \qquad (9)$$

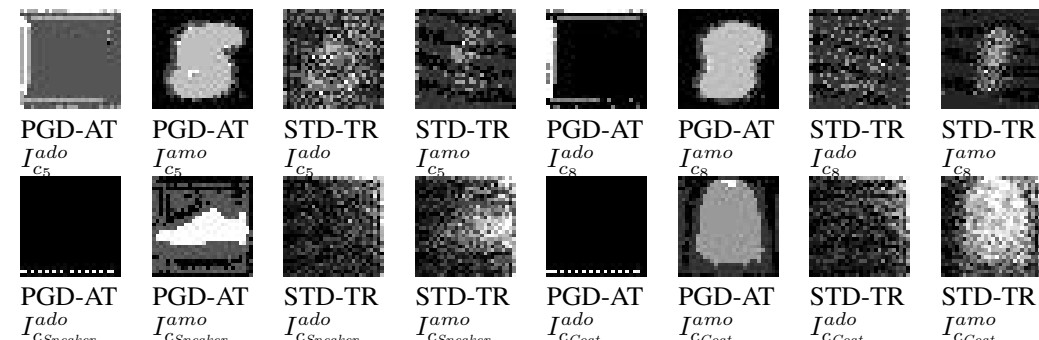

Figure 10: Image $I$ which triggers dominating active gating pattern per class over CONV DLGN-N128-D4 model obtained on adversarial examples alone but not on original examples (odd columns) and obtained both on original examples and adversarial examples(even columns). Column 1,2,5,6 is on PGD-AT model and column 3,4,7,8 is on STD-TR models.Loss function is as per Equation (7),$\lambda = 0.9, \alpha = 0.1$,optimization is as per Equation (8).For brevity, we have reported few class results. Detailed results are in Table 9.

We report visualized images $I_c^{ado}, I_c^{amo}$ for both DLGN PGD-AT and STD-TR models as before trained on MNIST, Fashion MNIST dataset in Figure 10. In the PGD-AT model, $I_c^{ado}$ does not produce meaningful inputs, as these patterns are framed images with no resemblance to any class, even using the same visualization method and parameters. This contrasts with $I_c^{adv}$, where adversarial examples resemble class images, indicating that only the dominant active gating patterns from original examples are meaningful in PGD-AT models. In the STD-TR model, both $I_c^{adv}$ and $I_c^{ado}$ appear similar, with $I_c^{amo}$ showing little resemblance to class images, highlighting significant differences between active gates triggered by adversarial and original examples. In PGD-AT, $I_c^{amo}$ shows that adversarial examples trigger a subset of original class gating patterns, maintaining some class resemblance.

## 6 CONCLUSION AND FUTURE DIRECTIONS

In this work we utilized DLGN architectures to thoroughly study the difference in properties exhibited by PGD-AT and STD-TR models. We analyzed fully connected networks, focusing on properties such as hyperplane alignment, path-activity and found that PGD-AT models exhibit larger datapoint separation distances from hyperplanes, active pathways triggered during adversarial attacks in PGD-AT models show less overlap with original examples of other classes and less overlap among original samples of same class suggesting better capacity utilization. We examined convolutional networks to show that PGD-AT models reduce gating overlap among different classes during adversarial attacks. Additionally, we used visualization techniques to understand the dominant gating patterns triggered per class in various scenarios for both STD-TR and PGD-AT models shedding light on the nature of representations used by these models.

We believe that leveraging the results of our analysis to develop novel algorithms that account for the properties examined could effectively enhance robustness. Extending this analysis to larger and more complex models, such as transformers or other deep architectures, could provide further insights into the generalizability of our findings. While this work focused on PGD-AT, other adversarial training methods could be explored to generalize the understanding of robustness.

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

# A APPENDIX

Network architecture of DLGN is shown in Figure 11.

PGD-40 and clean accuracies over MNIST and Fashion MNIST dataset using various architectures are reported at Table 3.

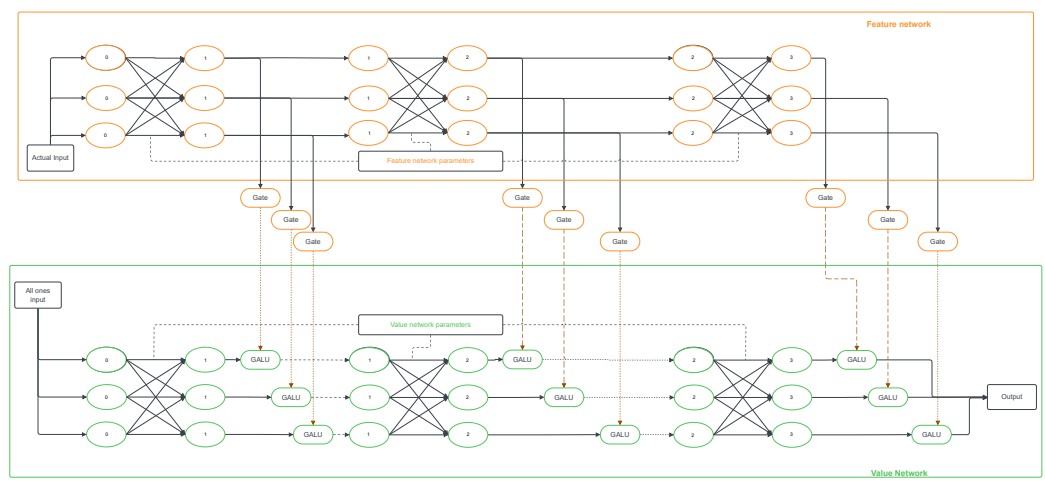

Figure 11: Deep Linearly Gated Networks (DLGN) network architecture. $GALU = x * Gate(x^`)$

| Dataset | Architecture | Training Type | PGD-40 Test Acc. ($@\epsilon = 0.3, @\epsilon = 0.2$) | Clean Test Acc. |
|---------|-------------|---------------|-------------------------------------------------------|-----------------|
| MNIST | FC-DLGN-W128-D4 | PGD-AT | (49.8%,54.5%) | 66.4% |
| | FC-DLGN-W128-D4 | STD-TR | (1.6%,2.7%) | 97.7% |
| | CONV-DLGN-N128-D4 | PGD-AT | (78.9%,88.9%) | 97.5% |
| | CONV-DLGN-N128-D4 | STD-TR | (0.05%,0.06%) | 98.4% |
| Fashion MNIST | FC-DLGN-W128-D4 | PGD-AT | (40.2%,48.3%) | 62.6% |
| | FC-DLGN-W128-D4 | STD-TR | (3.7%,5.1%) | 88.6% |
| | CONV-DLGN-N128-D4 | PGD-AT | (49.8%,88.9%) | 67.8% |
| | CONV-DLGN-N128-D4 | STD-TR | (0%,0%) | 88.9% |

Table 3: PGD-AT vs STD-TR model PGD accuracies and clean accuracies

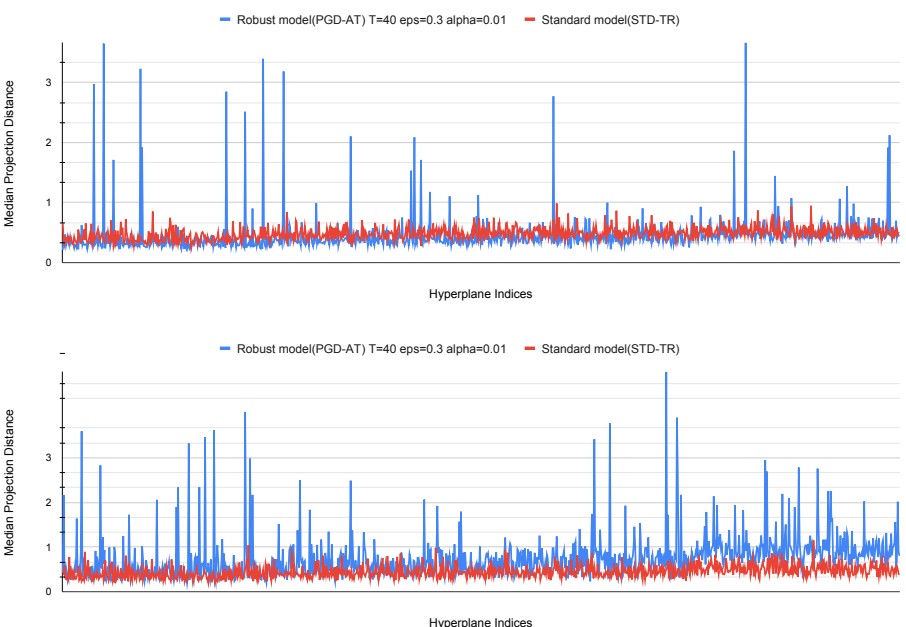

Figure 12: PGD-AT vs STD-TR FC-DLGN-W256-D4 median projection distance. The top image is on MNIST, and the bottom image is on the Fashion MNIST dataset. The Y-axis denotes the median projection distance of data points at node/hyperplane indices on the X-axis.

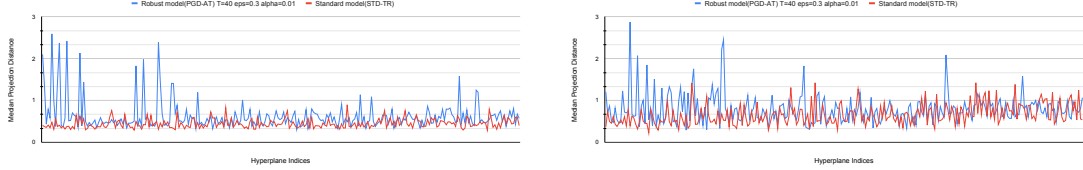

Figure 13: PGD-AT vs STD-TR FC-DLGN-W64-D4 median projection distance. The left image is on MNIST, and the right image is on the Fashion MNIST dataset. The Y-axis denotes the median projection distance of data points at node/hyperplane indices on the X-axis.

### A.1 MORE ANALYSIS OF HYPERPLANES IN FEATURE NETWORK OF PGD-AT AND STD-TR MODELS

The median projection distance at each hyperplane in PGD-AT and STD-TR models of DLGN with width 256 (see Figure 12) and 64 (see Figure 13) also clearly shows that median distances increase in robust models.

The projection distance histogram at hyperplanes, which shows significant differences in median projection distance between standard and robust models (see Figure 14), also shows that the projection distance of datapoints is shifted to larger distances in PGD-AT than STD-TR models.

### A.2 HYPERPLANE ANALYSIS IN SYNTHETIC XOR DATASET

The synthetic XOR 2D dataset constructed with a gap from x=0.5 and y=0.5 axis is shown in Figure 15. The decision boundaries of PGD-AT models (see Figure 16) are closer to optimal compared to STD-TR (see Figure 17), ensuring that adversarial examples within $L_\infty$ bounds ($\epsilon = 0.3$) are correctly classified only by PGD-AT.

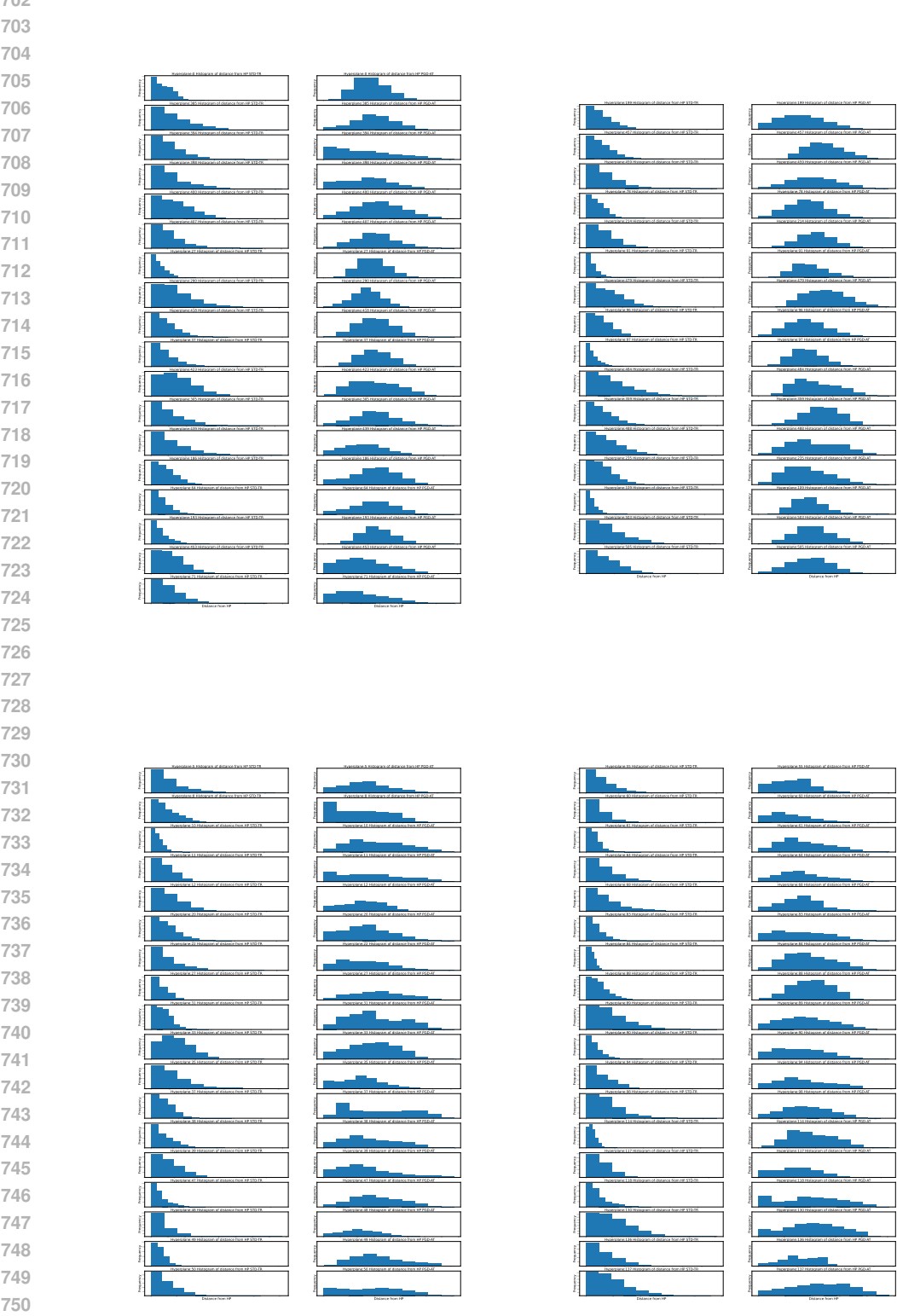

Figure 14: Projection distance distribution at hyperplanes whose medians differ significantly(by 0.5) between standard and robust DLGN models. Each row in each image denotes a hyperplane, with the Y-axis indicating the frequency of occurrence and the X-axis being the distance from that row's hyperplane. Columns 1,3 are for the STD-TR model, and columns 2,4 are for the PGD-AT model. Both X & Y axis is shared per row. First-row images correspond to the MNIST dataset, and the second-row images correspond to the Fashion MNIST dataset.

.

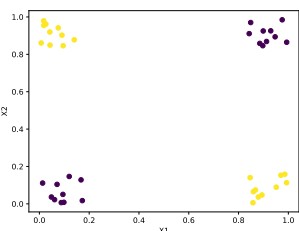

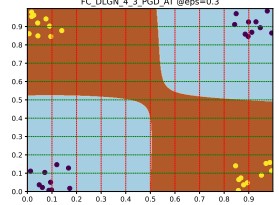

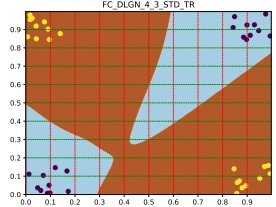

Figure 15: 2D XOR dataset with gap from x=0.5,y=0.5 being 0.32 to facilitate PGD-AT with eps < 0.32

Figure 16: PGD-AT decision boundary

Figure 17: STD-TR decision boundary

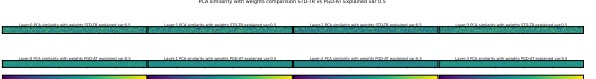

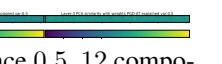

MNIST dataset with explained variance 0.5, 12 components.

Fashion MNIST dataset with explained variance 0.75, 15 components.

Figure 18: Effective weights with top PCA components in PGD-AT(bottom row) and STD-TR(top row) using FC-DLGN-W256-D4 architecture.

### A.3 PCA ANALYSIS IN ROBUST AND STANDARD MODELS

We report similarity of principal components with hyperplanes of feature network of DLGN with width 256 in Figure 18 and width 64 in Figure 19 respectively.

### A.4 MORE RESULTS IN ACTIVE SUBNETWORK OVERLAP IN PGD-AT VS STD-TR MODELS

The subnetwork overlap metrics for FC-DLGN _W128_D4 architecture trained over the Fashion MNIST dataset is shown in Table 4.

### A.5 QUALITATIVE ANALYSIS OF GATING PATTERNS IN PGD-AT AND STD-TR MODELS

We qualitatively inspect the difference in active gate counts with and without attacks using $\Lambda_c^{adv\_diff\_org}$ in Equation (10b) that measures the difference in active gate count for adversarial and original examples and is plotted per class for both PGD-AT and STD-TR models as an image of size $L * C_l, W, H$ in Table 5 and Table 6.

$$\Lambda_c^{mode} = \sum_{i=1}^{N_c} Gate(F^{mode}(X_c)), \qquad \in R^{L,C_l,W,H} \qquad (10a)$$

where $mode$ is either original examples or adversarial examples

$$\Lambda_c^{adv\_diff\_org}(i) = \Lambda_c^{adv}(i) - \Lambda_c^{org}(i), \qquad \forall\, i \in R^{L,C_l,W,H} \qquad (10b)$$

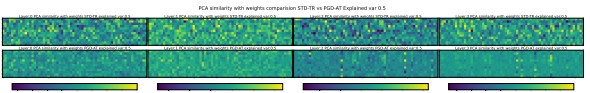

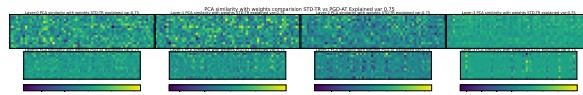

MNIST dataset with explained variance 0.5, 12 components.

Fashion MNIST dataset with explained variance 0.75, 15 components.

Figure 19: Effective weights with top PCA components in PGD-AT(bottom row) and STD-TR(top row) using FC-DLGN-W64-D4 architecture.

| Dataset | Train Type | PGD-40 Acc. | Clean Acc. | $log_2$ $\Psi^D_{orig}$ | $log_2$ $\Psi^S_{orig}$ | $log_2$ $\Psi^D_{adv}$ | $log_2$ $\Psi^S_{adv}$ | $log_2$ $\Psi^D_{adv,or}$ | $log_2$ $\Psi^S_{adv,or}$ |
|---------|-----------|-------------|-----------|-----------|-----------|-----------|-----------|-----------|-----------|
| FaMNIST 1vs9 | PGD-AT | 93.90% | 99.70% | 22.78 | 30.45 | **28.60** | **31.01** | **26.43** | **29.81** |
|  | STD-TR | 0.00% | 100.00% | 28.37 | 31.31 | 31.25 | 32.41 | 31.26 | 30.01 |
| FaMNIST 3vs8 | PGD-AT | 76.75% | 90.45% | **25.92** | **29.60** | **28.58** | **29.29** | **27.53** | **28.74** |
|  | STD-TR | 4.65% | 99.30% | 25.88 | 30.39 | 30.41 | 31.68 | 29.04 | 30.01 |
| FaMNIST 7vs9 | PGD-AT | 80.75% | 87.30% | **26.28** | **29.05** | **28.29** | **28.98** | **27.38** | **28.54** |
|  | STD-TR | 0.00% | 97.00% | 26.58 | 29.47 | 31.27 | 31.67 | 29.48 | 29.61 |
| FaMNIST 0vs2 | PGD-AT | 74.25% | 90.10% | **26.68** | **30.04** | **27.61** | **29.00** | **27.06** | **29.01** |
|  | STD-TR | 0.00% | 97.10% | 28.74 | 30.79 | 30.10 | 32.15 | 30.73 | 29.17 |
| FaMNIST 4vs5 | PGD-AT | 92.75% | 98.90% | **22.70** | **29.82** | **29.87** | **30.24** | **28.87** | **29.09** |
|  | STD-TR | 23.80% | 99.00% | 27.88 | 31.16 | 31.31 | 31.91 | 30.55 | 30.08 |
| FaMNIST 6vs7 | PGD-AT | 89.00% | 98.40% | **23.36** | **29.98** | **28.47** | **31.02** | **31.32** | **30.30** |
|  | STD-TR | 23.80% | 100.00% | 27.35 | 31.50 | 23.50 | 29.21 | 27.44 | 28.73 |

Table 4: FC-DLGN-W128-D4 architecture PGD-AT vs STD-TR model path overlaps metrics over original and adversarial examples. The task is binary classification over the Fashion MNIST dataset in column 2, and the model has a single output node for classification. PGD-AT rows are highlighted in bold for better readability.

## A.6 QUANTITATIVE ANALYSIS OF GATING PATTERNS IN PGD-AT AND STD-TR MODELS

The $IOU^{org}_{agc}$, $IOU^{adv}_{agc}$ is measured for each pair of classes in Table 7 for Fashion MNIST dataset.

## A.7 INTERPRETATION OF GATING PATTERNS IN PGD-AT VS STD-TR MODELS

The visualizations ($I^{org}$, $I^{adv}$) for CONV-DLGN _N128_D4 trained on the MNIST, Fashion MNIST dataset are presented in Table 8. We report visualized images $I^{ado}_c$, $I^{amo}_c$ for both CONV-DLGN _N128_D4 PGD-AT and STD-TR models as before trained on MNIST, Fashion MNIST dataset in Table 9.

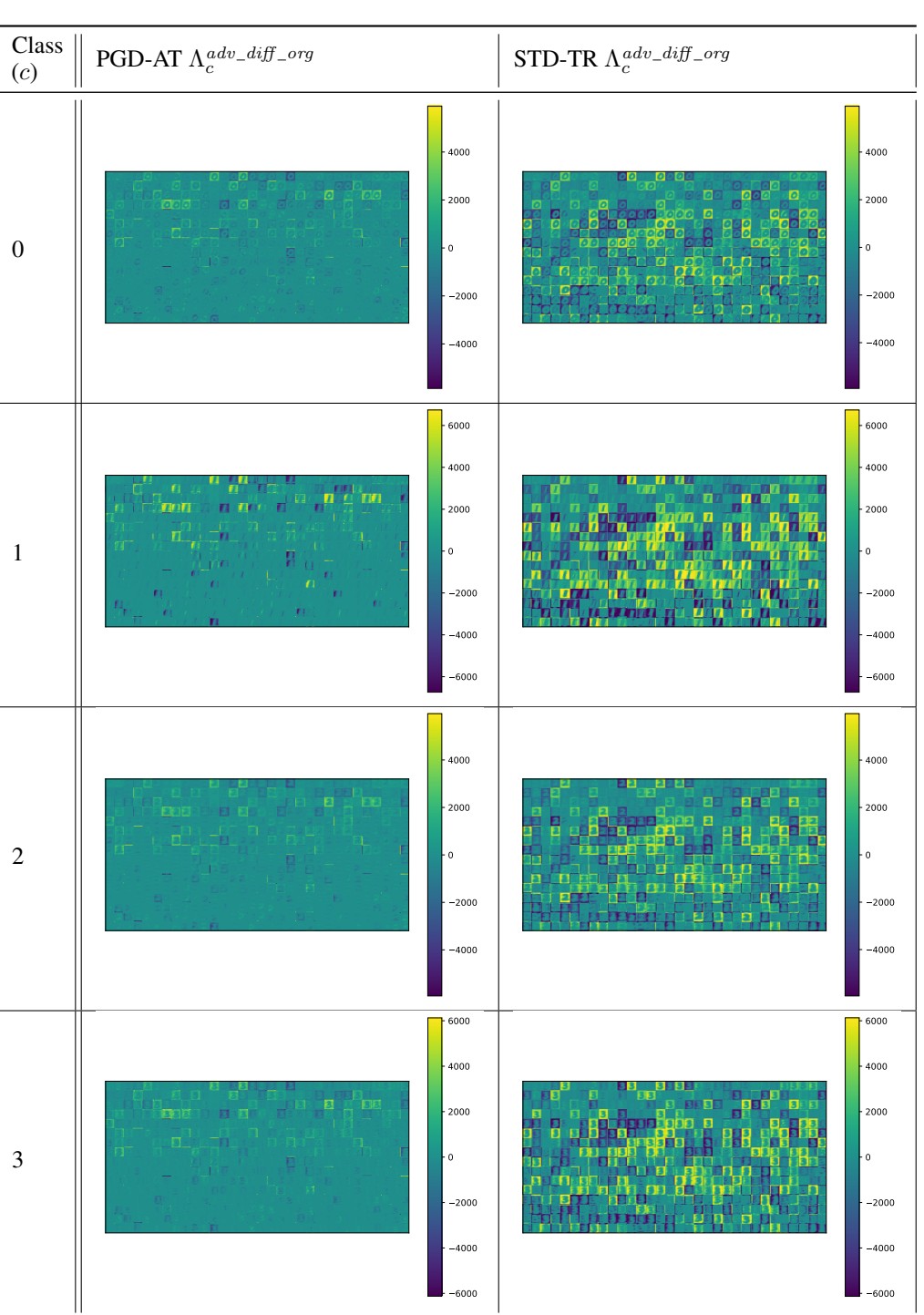

Table 5: $\Lambda_c^{adv\_diff\_org}$ for PGD-AT and STD-TR models with CONV_N128_D4 DLGN architecture on MNIST dataset. In each cell of the image, every four rows represent a layer's $\Lambda_{l,c}^{adv\_diff\_org}$

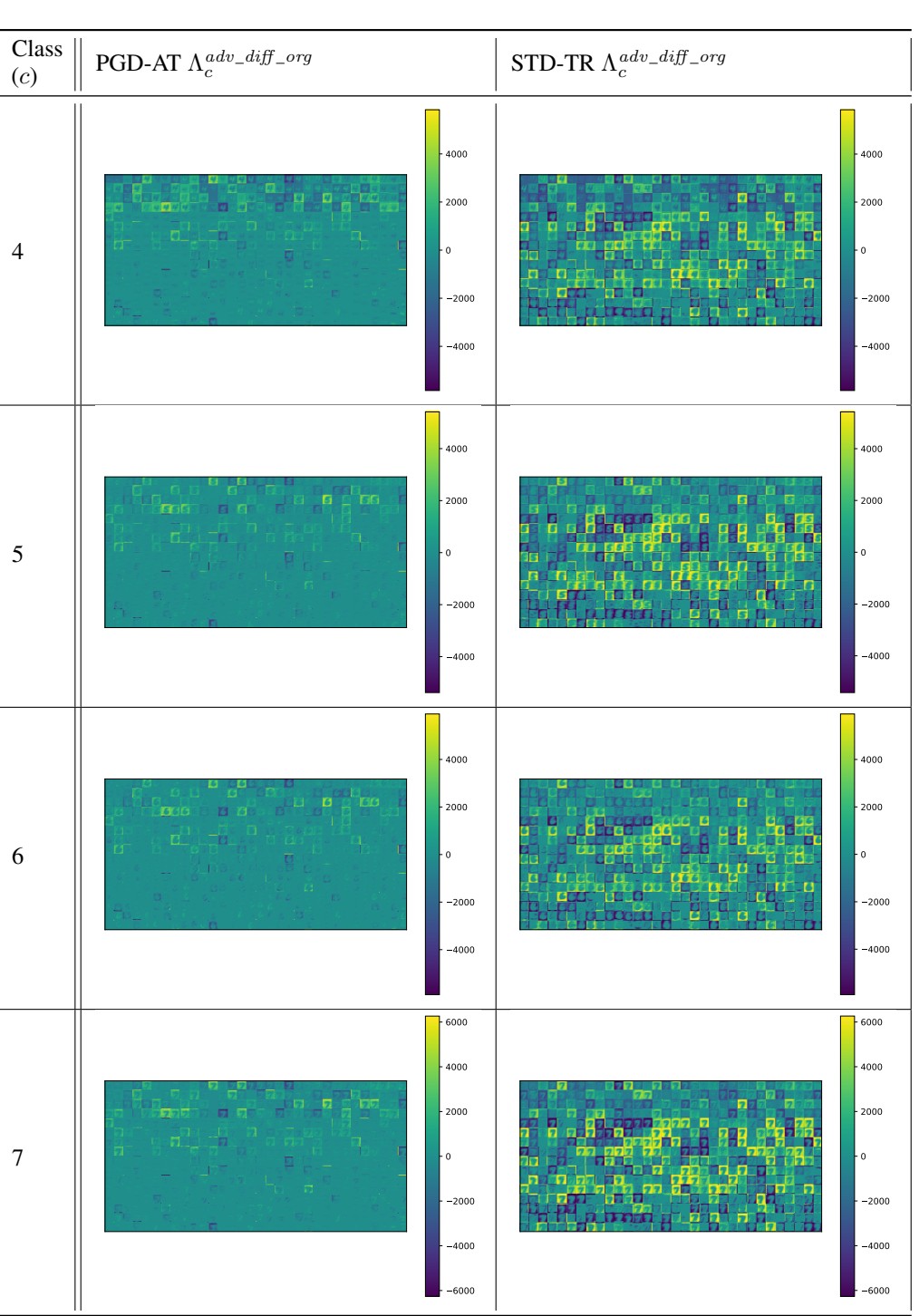

Table 6: $\Lambda_c^{adv\_diff\_org}$ for PGD-AT and STD-TR models with CONV_N128_D4 DLGN architecture on MNIST dataset. In each cell of the image, every four rows represent a layer's $\Lambda_{l,c}^{adv\_diff\_org}$

| Src Class | Train Type | Quantity | Class 0 | Class 1 | Class 2 | Class 3 | Class 4 | Class 5 | Class 6 | Class 7 | Class 8 | Class 9 |
|---|---|---|---|---|---|---|---|---|---|---|---|---|
| 0 | PGD-AT | $IOU_{agc}^{adv}$ | 100.0 | 81.0 | 86.9 | 86.4 | 86.0 | 75.0 | 89.6 | 73.5 | 80.9 | 75.6 |
|  |  | $IOU_{agc}^{org}$ | 100.0 | 75.6 | 84.3 | 83.0 | 80.9 | 68.1 | 86.5 | 67.4 | 74.9 | 69.4 |
|  | STD-TR | $IOU_{agc}^{adv}$ | 100.0 | 72.1 | 78.6 | 79.4 | 76.9 | 67.9 | 83.2 | 65.3 | 73.9 | 67.7 |
|  |  | $IOU_{agc}^{org}$ | 100.0 | 57.1 | 69.2 | 69.9 | 65.4 | 47.3 | 76.6 | 44.4 | 59.6 | 49.4 |
| 1 | PGD-AT | $IOU_{agc}^{adv}$ | 81.0 | 100.0 | 77.8 | 88.2 | 80.2 | 74.2 | 77.8 | 74.4 | 74.0 | 73.6 |
|  |  | $IOU_{agc}^{org}$ | 75.6 | 100.0 | 72.1 | 85.2 | 74.5 | 66.2 | 71.6 | 67.5 | 65.9 | 66.4 |
|  | STD-TR | $IOU_{agc}^{adv}$ | 72.1 | 100.0 | 68.5 | 82.4 | 71.1 | 64.9 | 70.9 | 65.5 | 66.7 | 64.2 |
|  |  | $IOU_{agc}^{org}$ | 57.1 | 100.0 | 51.0 | 72.8 | 55.1 | 45.9 | 52.1 | 44.9 | 46.8 | 45.4 |
| 2 | PGD-AT | $IOU_{agc}^{adv}$ | 86.9 | 77.8 | 100.0 | 82.3 | 91.3 | 76.0 | 93.1 | 73.5 | 84.8 | 77.5 |
|  |  | $IOU_{agc}^{org}$ | 84.3 | 72.1 | 100.0 | 77.4 | 89.8 | 69.2 | 91.4 | 67.2 | 78.8 | 72.6 |
|  | STD-TR | $IOU_{agc}^{adv}$ | 78.6 | 68.5 | 100.0 | 73.1 | 86.3 | 67.0 | 83.7 | 65.4 | 76.9 | 68.8 |
|  |  | $IOU_{agc}^{org}$ | 69.2 | 51.0 | 100.0 | 58.4 | 81.0 | 48.5 | 81.9 | 45.3 | 65.2 | 54.4 |
| 3 | PGD-AT | $IOU_{agc}^{adv}$ | 86.4 | 88.2 | 82.3 | 100.0 | 84.7 | 75.0 | 83.0 | 74.7 | 77.7 | 75.2 |
|  |  | $IOU_{agc}^{org}$ | 83.0 | 85.2 | 77.4 | 100.0 | 79.7 | 67.7 | 78.0 | 68.3 | 70.8 | 68.5 |
|  | STD-TR | $IOU_{agc}^{adv}$ | 79.4 | 82.4 | 73.1 | 100.0 | 75.4 | 68.4 | 77.5 | 68.2 | 71.6 | 68.3 |
|  |  | $IOU_{agc}^{org}$ | 69.9 | 72.8 | 58.4 | 100.0 | 62.2 | 49.4 | 62.8 | 47.2 | 54.7 | 49.4 |
| 4 | PGD-AT | $IOU_{agc}^{adv}$ | 86.0 | 80.2 | 91.3 | 84.7 | 100.0 | 76.1 | 91.3 | 74.4 | 84.1 | 78.2 |
|  |  | $IOU_{agc}^{org}$ | 80.9 | 74.5 | 89.8 | 79.7 | 100.0 | 68.0 | 87.9 | 67.1 | 78.0 | 72.5 |
|  | STD-TR | $IOU_{agc}^{adv}$ | 76.9 | 71.1 | 86.3 | 75.4 | 100.0 | 66.4 | 83.5 | 65.4 | 76.1 | 69.0 |
|  |  | $IOU_{agc}^{org}$ | 65.4 | 55.1 | 81.0 | 62.2 | 100.0 | 48.9 | 79.4 | 46.0 | 64.9 | 54.8 |

Table 7: CONV DLGN-N128-D4 PGD-AT vs STD-TR model IOU of active gate count between class-pairs over adversarial and original examples for Fashion MNIST dataset.

| MN Class ($c$) | PGD-AT $I^{org}$ | PGD-AT $I^{adv}$ | STD-TR $I^{org}$ | STD-TR $I^{adv}$ | Fashion MN ($c$) | PGD-AT $I^{org}$ | PGD-AT $I^{adv}$ | STD-TR $I^{org}$ | STD-TR $I^{adv}$ |
|---|---|---|---|---|---|---|---|---|---|
| 0 | | | | | Ankle-boot | | | | |
| 1 | | | | | Bag | | | | |
| 2 | | | | | Coat | | | | |
| 3 | | | | | Dress | | | | |
| 4 | | | | | Pullover | | | | |
| 5 | | | | | Sandal | | | | |
| 6 | | | | | Shirt | | | | |
| 7 | | | | | Sneaker | | | | |
| 8 | | | | | T-shirt | | | | |
| 9 | | | | | Trouser | | | | |

Table 8: Image $I$ which triggers dominating gating pattern per class obtained on adversarial examples (column 3,5,8,10) and original examples (column 2,4,7,9). Columns 2,3,7,8 are on the PGD-AT model, and columns 4,5,9,10 are on the STD-TR model. Loss function is as per Equation (7),$\lambda = 0.9, \alpha = 0.1$,optimization is as per Equation (8)

| MN Class (c) | PGD-AT $I^{ado}$ | PGD-AT $I^{amo}$ | STD-TR $I^{ado}$ | STD-TR $I^{amo}$ | Fashion MN (c) | PGD-AT $I^{ado}$ | PGD-AT $I^{amo}$ | STD-TR $I^{ado}$ | STD-TR $I^{amo}$ |
|---|---|---|---|---|---|---|---|---|---|
| 0 | | | | | Ankle-boot | | | | |
| 1 | | | | | Bag | | | | |
| 2 | | | | | Coat | | | | |
| 3 | | | | | Dress | | | | |
| 4 | | | | | Pullover | | | | |
| 5 | | | | | Sandal | | | | |
| 6 | | | | | Shirt | | | | |
| 7 | | | | | Sneaker | | | | |
| 8 | | | | | T-shirt | | | | |
| 9 | | | | | Trouser | | | | |

Table 9: Image $I$ which triggers dominating active gating pattern per class obtained on adversarial examples alone but not on original examples (columns 2,4,7,9) and obtained both on original examples and adversarial examples(columns 3,5,8,10). Columns 2,3,7,8 are on the PGD-AT model, and columns 4,5,9,10 are on the STD-TR models. Loss function is as per Equation (7),$\lambda = 0.9, \alpha = 0.1$,optimization is as per Equation (8)