# OpenReview forum: "Interpreting Adversarial Attacks and Defenses using Architectures with Enhanced Interpretability"
_ICLR.cc/2025/Conference — Submitted to ICLR 2025_

### Official Review · Reviewer_aRGv · 2024-10-18

**Soundness:** 2
**Presentation:** 1
**Contribution:** 2
**Rating:** 3
**Confidence:** 3

**Summary:**

This paper uses the Deep Linearly Gated Network (DLGN) to enhance interpretability in AT models compared to standard training. The paper aims to understand how robust models differ from standard ones by analyzing properties such as hyperplane alignment, active subnetwork overlap, gating patterns, and principal component similarity. The authors demonstrate that AT-based models exhibit less active subnetwork overlap, larger distances from hyperplanes, and more distinct gating patterns, leading to improved robustness.

**Strengths:**

The paper provides many experiments to discover the difference between standard and AT models, including analyses of hyperplane distance, PCA alignment, and gating patterns.

**Weaknesses:**

The paper is poorly written and is very hard to follow. Many notations and equations are not clear. The presentation also needs to improve, like the citation format is inconsistent, and the Figure 8 is beyond the margin.

The paper uses Deep Linearly Gated Network (DLGN) to analyze. This network seems impractical as it doesn’t have activation functions for feature layers.

The experiments use simple datasets (MNIST, Fashion MNIST), which may not generalize well to more complex scenarios. At least testing on CIFAR-10.

**Questions:**

Can you provide some advantages of using DLGN as classifier?

Can your analyses be generalized to more AT-based techniques, like TRADES and MART?

Could your analyses instruct a new algorithm to improve the robustness?

---

### Official Review · Reviewer_tA1a · 2024-10-30

**Soundness:** 2
**Presentation:** 2
**Contribution:** 1
**Rating:** 3
**Confidence:** 4

**Summary:**

This paper proposes to use Deep Linearly Gated Networks (DLGN)to analyze adversarial robustness rather than regular network architectures because of the better interpretation capabilities. This paper uses feature networks in DLGN with fully connected layers or CNN layers to qualitatively and quantitatively contrast gating patterns between robust and standard models.

**Strengths:**

- Studying an important topic about interpreting adversarial attacks.
- Proposing a very promising topic about using Deep Linearly Gated Networks to analyze adversarial robustness.

**Weaknesses:**

- The finding, ''Our analyses show that hyperplanes in PGD-AT (FC) models are farther from data points compared to STD-TR (FC) models and play a key role in enhancing robustness.'' is not surprising. This has been found in existing works[1,2] by using T-SNE plots.

- There are many papers that analyzed adversarial training and adversarial attacks with theoretical proof.

- The results are specific to a particular architecture. It is unknown whether it still holds in other larger architectures and larger datasets.

- The presentation of results is confusing to me. I expect the authors to utilize their new findings to improve the current adversarial training. In other words, the new findings should also be validated by incorporating existing methods. If the new findings are correct, we should see improvements.

[1] Distilling Robust and Non-Robust Features in Adversarial Examples by Information Bottleneck

[2] IB-RAR: Information Bottleneck as Regularizer for Adversarial Robustness

**Questions:**

How do we use the new findings in the paper to further develop better methods?

Are the new findings validated on larger datasets and architectures?

---

### Official Review · Reviewer_2BKD · 2024-11-04

**Soundness:** 3
**Presentation:** 1
**Contribution:** 2
**Rating:** 3
**Confidence:** 4

**Summary:**

This paper uses Deep Linearly Gated Networks to study differences between adversarially trained and standard trained neural networks particularly with regard to the position of the data relative to the hyperplanes of the linear layers, the subnetworks activated by natural and attacked inputs, and gating patterns induced by clean and perturbed data. The key contributions of this paper include (1) showing that the hyperplanes of linear layers lie further from the data points in adversarially trained models than in standard trained models, (2) active subnetworks for different classes do not overlap in adversarially trained models and (3) gate overlap (the overlap between activation patterns of intermediate neurons when stimuli from different classes are presented) increases more for standard trained models than adversarially trained models.

**Strengths:**

1. Using Deep Linearly Gated Networks (DLGN) for studying and interpreting adversarial attacks is, to the best of my knowledge, novel.
1. The observation that adversarially trained models exhibit higher gate overlap on clean images is an interesting finding.

**Weaknesses:**

1. The significance of this study is unclear. Currently, the paper simply contains a series of experimental observations however these observations, in themselves, are not useful for other researchers and practitioners because (1) there is no evidence to indicate that these observations would also apply to models and datasets that are generally used in practice and (2) there is no actionable recommendation or insight provided that others can use to make models more adversarially robust. I encourage the authors to provide more details and discussions in this regard because otherwise this study will have very limited impact.
   1. MNIST and Fashion-MNIST are too simple to make any generalizable conclusions. The literature on adversarial robustness is replete with results that hold only on (F)MNIST but not on other datasets (e.g. [1]). I strongly suggest that the authors include more realistic datasets (at least CIFAR-10) in their analysis.

1. The presentation lacks clarity and the paper is rather difficult to read and understand. Below are the most significant issues I encountered:
   1. The first section is poorly written. It starts by introducing and motivating DLGNs and then provides background of adversarial attacks before briefly mentioning in 3.5 lines what the current study is about. This leaves the reader confused and unaware of what the authors are presenting in the paper, which makes the remainder of the paper hard to read. I recommend starting with a brief overview of the study's goals before providing background on DLGNs and adversarial attacks.
   1. I find the practice of placing the **Notations** section _before_ the section in which the notations are used to be very confusing. I strongly suggest moving that notations within the relevant section
   1. The concept of paths and nodes is not well explained in section 1 so it is not clear what the path indexing used in equation 1b means.
   1. The text in the figures is too small making it near impossible to read the figures because zooming in enough to read the text usually leads to the figure being cropped (at least on my screed).
   1. The 2D XOR task is not properly introduced in section 2.2. Please provide details about the task.
      1. Figures 2 and 3 can be combined
      1. Figure 5 has too many lines and thus is very hard to read. I would recommend splitting it into multiple figures or using a different type of visualization.
      1. The red and green lines in Figure 6 seem to only add clutter to the plot without conveying any information. It appears that there are some labels for them but I can't read them even that 5x zoom.

1. Some of the results are rather expected.
   1. It is expected that adversarial training will lead to hyperplanes further from the data points because during adversarial training each data point is effectively transformed into a _ball_ around the original data point. By training the model to respond similarly to every point in the ball we are effectively pushing the hyperplanes away from the data point by the distance of at least the radius of the ball.
   1. It is known that adversarially trained models exhibit higher specificity of their activation patterns and thus inverting activations (gating patterns) results in images closer to the original image [2]



[1] Paiton, Dylan M., et al. "Selectivity and robustness of sparse coding networks." Journal of vision 20.12 (2020): 10-10.

[2] Feather, Jenelle, et al. "Model metamers illuminate divergences between biological and artificial neural networks." bioRxiv (2022): 2022-05.

**Questions:**

1. It is not clear why DLGNs are required for the analysis being performed in this work. Would the same analysis not be possible with standard ReLU networks?
1. $\hat{y}$ seems to be a scalar. Shouldn't it be a vector for multi-class classification?
1. Why is the median distance being compared in section 2.1? Would it not be more appropriate to do a statistical test to determine if the hyperplane distances for the AT model are greater than the ST model?
1. The various model configurations that were used have not been adequately justified. For example, for the 2D XOR the minimal architecture would be 1 hidden layer with 3 ReLU neurons, but a much larger model is used in the paper. Please provide some discussion on why the particular number of neurons and layers are used and would the results persist across different architectures.

---

### Meta-Review · Area_Chair_NzpD · 2024-12-20

**Metareview:**

This paper uses Deep Linearly Gated Networks (DLGN) to analyze the adversarial robustness and claims that it can achieve better interpretability. All the reviewers think that the paper is poorly written and hard to read. The findings are obtained from experiments on small datasets and for a particular architecture and, therefore, may not be very useful.

**Additional Comments On Reviewer Discussion:**

The authors did not provide a rebuttal to address the reviewers' questions and concerns.

---

### Decision · Program_Chairs · 2025-01-22

Reject